# Frailty and Quality of Life among Older Adults in Communities: The Mediation Effects of Daily Physical Activity and Healthy Life Self-Efficacy

**DOI:** 10.3390/geriatrics7060125

**Published:** 2022-11-05

**Authors:** Chia-Hui Lin, Chieh-Yu Liu, Chun-Ching Huang, Jiin-Ru Rong

**Affiliations:** 1School of Nursing, Chang Gung University of Science and Technology, Chang Gung Medical Foundation, Chiayi 613016, Taiwan; 2Biostatistical Consultant Laboratory, Department of Health Care Management, National Taipei University of Nursing and Health Sciences, Taipei 112, Taiwan; 3School of Nursing, National Taipei University of Nursing and Health Sciences, Taipei 112, Taiwan

**Keywords:** frailty, quality of life, daily physical activity, healthy life self-efficacy, serial multiple mediation model

## Abstract

As the global population ages, frailty, which has been shown to affect and predict the quality of life (QoL) of older adults, has become a central issue. The aim of this study was to explore the mediating effects of daily physical activity (DPA) and healthy life self-efficacy (HLSE) on the relationship between frailty and QoL in older adults using a serial multiple mediation model. The cross-sectional study was conducted among 210 community-dwelling older adults in Taiwan. Data were collected using the Taiwanese version of the Tilburg Frailty Indicator, the EuroQoL visual analog scale, the Kihon Checklist, and the Chronic Disease Self-Efficacy Scales. The PROCESS macro for SPSS based on the bootstrap method was used to determine the mediating effects of DPA and HLSE on the relationship between frailty and QoL. The results showed that frailty was found to have both direct and indirect effects on QoL. As predicted, DPA and HLSE partially mediated the relationship between frailty and quality of life (DPA: B = −0.71, *p* < 0.001; HLSE: B = −0.32, *p* < 0.001). In addition, serial mediation analyses indicated that the association between frailty and QoL was partially mediated by DPA and HLSE in a sequential manner (B = −0.16, *p* < 0.001). The serial mediation has a causal chain linking DPA and HLSE, with a specified direction of causal flow. According to the results of the serial multiple mediation model, the elderly should be encouraged to continue their activities in daily life, which not only improves self-efficacy and confidence in maintaining health but also reduces the negative impact of frailty on QoL.

## 1. Introduction

Older adults are affected by aging and chronic diseases, which increase the incidence of frailty problems. Aging and frailty reduce older adults’ ability to maintain healthy activity levels, lower their self-efficacy, and even affect their quality of life (QoL) [1,2,3]. For community-dwelling older adults, QoL is a subjective appraisal of the condition of an individual’s general function [4]. The changes that come with aging, disease, and frailty affect their disability status, dependence on family members, and burden on society [5,6]. Frailty has also been found to cause hospitalization and death [5,7,8]. How to reduce the negative impact of frailty on QoL should thus be explored.

Researchers have found that physical activity can mediate or moderate the influence of frailty on healthy living among older adults [9,10]. Previous studies have also found that physical activity and self-efficacy have a significantly positive effect on QoL [11]. Physical activity includes physical exercise and activities related to daily life, such as instrumental activities of daily living [12]. According to caregivers for older adults and their relevant professional experience, most elderly people are not interested in physical exercise or extreme physical activities, although they may still have the ability to perform their daily activities. There have been few studies that have directly addressed the relationship between daily functioning and QoL in older adults.

Within the aging framework, activity theory states that a person’s activity level is positively related to their life satisfaction [13]. The ability to perform daily activities is essential for self-care; if older adults develop severe impairments in performing self-care activities, they may become dysfunctional or disabled, which are conditions that are highly related to lower QoL [12]. Some studies have reported that impairments in daily activity are significantly associated with frailty [14,15]. Using the activity theory of aging and findings from previous studies, we constructed a mediation model to test the role of daily physical activity (DPA) in the relationship between frailty and QoL in older adults.

In the social cognitive theory framework proposed by Albert Bandura (1986), self-efficacy is defined as a person’s belief in their capability to successfully perform a particular task [16]. Self-efficacy can enhance a person’s motivation and confidence in performing healthy behaviors and can increase personal satisfaction and participation in healthy activities [17]. In fact, self-efficacy serves an essential mediating function in personal health. Health-related self-efficacy can affect an individual’s ability to cope with different health problems and exhibit confidence in maintaining their health. Research has shown that self-efficacy is correlated with self-control behaviors and positively impacts QoL [18]. An individual with low perceived self-efficacy, or low confidence in controlling their health, also has a low QoL [19]. In addition, a Korean study supporting the notion that self-efficacy plays a mediatory role between frailty and health-related quality of life was conducted by Choi and Jeon [20]. Using the social cognitive theory and the findings from previous studies, we constructed a mediation model to test the role of healthy life self-efficacy (HLSE) in the relationship between frailty and QoL in older adults.

The health promotion model (HPM) states that each person has unique personal characteristics and experiences that affect health actions and behavioral outcomes [21]. Studies applying the health promotion model in order to change one’s unhealthy behaviors and promote health outcomes commonly focus on adults [21,22]. The major concepts that construct the HPM are individual characteristics and experiences (such as prior related behavior), behavior-specific cognitions and effects (such as perceived self-efficacy), and behavioral outcomes (such as health-promoting behavior). In accordance with the HPM, this study explored the relationship between frailty, DPA, HLSE, and QoL. The major research concepts are daily physical activity (as prior relevant behaviors), self-efficacy for healthy living (as cognitions and effects of specific behaviors), and quality of life (as health promotion behavioral outcomes).

Physical activity plays a role in predicting an individual’s self-efficacy toward exercise [9,10]. Self-efficacy has been shown to support individuals in performing healthy behaviors and the belief that they can control their own health [23]. If elderly people have sufficient health-related self-efficacy and confidence, they can also improve their health behaviors and QoL [19]. Studies have found that physical activity does not have to be vigorous to have benefit [24]. However, if older adults can maintain DPA, they may develop good confidence and self-efficacy toward maintaining and improving their QoL. In addition, HPM constructs could function as a causal chain linking the mediators with a specified direction of causal flow, a model referred to as serial mediation [25].

Previous studies have explored the independent contributions of these factors, but no studies have explored all these factors (frailty, DPA, HLSE, and QoL) in combination. The aim of this research is to determine the mediating effects of DPA and HLSE on the relationship between frailty and QoL in older adults using a serial multiple mediation model. Therefore, we aimed to propose a sequential mediation model to further analyze the relationship between frailty, daily physical activity, self-efficacy, and quality of life. This model can be used to elucidate relevant mechanisms, prevent frailty, and improve quality of life in community-based older adults.

## 2. Design and Methods

This study used a descriptive cross-sectional design to evaluate whether the effect of frailty on QoL is mediated by DPA and HLSE in older adults with a serial multiple mediation model. This model can help us understand how frailty and QoL are linked, explain any sequential causality among the two mediators (DPA and HLSE), and aid the investigation into whether frailty has direct or indirect effects on QoL.

A sample of 210 community-dwelling older adults (aged over 60) from two rural counties (Chiayi and Yunlin) in Taiwan was selected. The participants were recruited between 15 April 2020, and 30 March 2021, using convenience sampling. The eligibility criteria for participation included the ability to engage in conscious and coherent verbal communication with the interviewer and being over 60 years of age. The exclusion criteria were diagnosis with a mental disorder, drug, or alcohol addiction, severe visual or hearing impairment, and refusal to participate.

To confirm the adequacy of the sample size for regression analysis, we used G*Power (version 3.1.9.4) as designed by Faul et al. [26]. The number of predictors was set to five, and the other statistical parameters were set at their default values (a medium effect size of 0.15, α level of 0.05, and high power of 0.95). The a priori sample size was computed to be 129, which is lower than the actual sample size of this study (*N* = 210). This study was approved by the institutional review board (IRB) committee of a medical institution (ref. 02-012). After obtaining written informed consent, a questionnaire survey was conducted with an average completion time of 30–40 min.

## 3. Measures

The measurement tools of this study all have good reliability and validity.

### 3.1. Frailty

Frailty was measured by the Taiwanese version of the Tilburg Frailty Indicator (TFI-T) [27]. The scale is a widely used frailty screening scale with good reliability and validity [27] that consists of 15 questions, resulting in a total frailty score ranging from 0 to 15. A higher total score indicates a worse degree of frailty. Lin’s (2021) study confirms that a TFI-T tool measurement score of 5.5 can be used as a cutoff value for the degree of frailty [27]. Frailty was defined based on three components: physical, psychological, and social frailty [28].

### 3.2. Quality of Life (QoL)

QoL was assessed using the EuroQoL visual analog scale (EQ-VAS), which is a 20 cm vertical analog scale with scores ranging from 0 to 100 points; higher scores indicate higher quality of life: 0 represents “the worst health you can imagine” and 100 represents “the best health you can imagine” [29]. Respondents directly marked a line on the scale with the corresponding score representing their health status on the day of the interview [30]. The EQ-VAS questionnaire is cognitively undemanding, taking only a few minutes to complete. We used the EQ-VAS as the measure of overall self-rated health status and as the dependent variable of this research. There is extensive literature supporting its validity and reliability [31,32]. The EQ-VAS has been demonstrated to be a reliable and valid measure for assessing quality of life in the Taiwanese population [33,34].

### 3.3. Daily Physical Activity (DPA)

DPA was measured as the ability to perform daily activities and was extracted from the two corresponding parts of the Kihon Checklist [35]. The first part involves IADL measurement (five items: transportation ability, ability to buy groceries or clothing, ability to manage money, ability to visit friends, and ability to communicate with friends and family), and the second part assesses physical strength (four items: ability to stand up from a chair, ability to walk up and down stairs unaided, ability to walk continuously for more than 15 min, and whether they have fallen during the past year). These measures help clarify an individual’s lifestyle and ability to perform daily activities. Each answer was dichotomous (yes or no) and there was a total of nine questions, with a resulting score ranging from 0 to 9; a higher score indicates a good level of daily activity.

### 3.4. Healthy Life Self-Efficacy

HLSE measures were adapted from the Chronic Disease Self-Efficacy Scales [36]. Participants indicated how confident they were in their ability to perform certain healthy activities. They responded to items such as “I am confident that I can eat regularly” or “I can exercise without the company of others”. The scores for these items related to managing health activities ranged from 1 (not at all confident) to 100 (totally confident).

## 4. Statistical Analysis

First, we calculated the descriptive statistics and bivariate correlations among variables of interest using SPSS v26.0 (IBM, Armonk, NY, USA) [37]. A two-tailed *p*-value less than 0.05 indicated statistical significance. According to Cohen (1992), a Pearson’s correlation coefficient around 0.10 indicates a minimum effect size, a coefficient near 0.30 indicates a moderate effect, and one above 0.50 indicates a strong effect [38].

Second, Model 6 was used for the serial multiple mediation analysis. The serial multiple mediation model proposed by Hayes was used to determine the mediating roles of DPA and HLSE on the relationship between frailty and QoL [39,40]. This model includes three indirect effects and one direct effect. The indirect effects are as follows: an indirect effect of frailty on QoL through DPA (M1, a1 × b1), an indirect effect of frailty on QoL through HLSE (M2, a2 × b2), and an indirect effect of frailty on QoL through the serial mediation of DPA and HLSE (M1M2, a1 × d21 × b2). The total effect of frailty (c) is the combination of the direct effect of frailty on QoL and all the indirect effects. Bootstrapping (5000 bootstrap samples) with a 95% confidence interval (CI) was conducted to test the significance of indirect effects [39,40]. The serial multiple mediation model was tested with the PROCESS V.3.4 macro for SPSS [40].

## 5. Results

### 5.1. Participant Characteristics

Among the 210 community-dwelling older adults sampled (range = 60−93 years old, mean age = 74.45 years, SD = 9.15), 75.70% were females. Of all the elderly individuals, 97% were married and 85% had a lower level of education (≤12 years). Descriptive analysis and differences in frailty scores, DPA, HLSE, and quality of life among participants with different demographic characteristics are illustrated in Table 1.

The mean TFI-T score was 5.69 (SD = 3.22). A cutoff score of ≥5.5 was used to diagnose frailty [27]. Accordingly, the conditions of the older adults in this study can be divided into two groups: frail (TFI-T ≥ 5.5, *n* = 109, 51.9%) and robust (TFI-T < 5.5, *n* = 101, 48.1%). These results show an even distribution of frail and robust adults. The mean HLSE score was 75.2 (SD = 20.76), the mean EQ-VAS score was 70.18 (SD = 15.64), and the mean DPA score was 6.65 (SD = 2.35) (Table 2). Considering the median score of 50 points for self-efficacy and QoL, the older adults in this study had medium-to-high levels of self-efficacy and QoL.

### 5.2. Correlations among Study Variables

A set of correlation analyses (Pearson or Spearman correlation coefficients) were run to test possible associations between frailty, DPA, HLSE, and QoL and sociodemographic features (age, gender, and education level). There was a positive correlation between QoL and HLSE (*r* = 0.51, *p* < 0.01), as well as between QoL and DPA (*r* = 0.54, *p* < 0.01). A positive correlation was also found between HLSE and DPA (*r* = 0.48, *p* < 0.01). There was a negative correlation between frailty and QoL (*r* = −0.59, *p* < 0.01), frailty and HLSE (*r* = −0.62, *p* < 0.01), and frailty and DPA (*r* = −0.57, *p* < 0.01). Age correlates positively with frailty (*r* = 0.37, *p* < 0.01). Females seem to be more likely to become frail (*r* = 0.18, *p* < 0.01). It was also interesting to note that educational level negatively correlated with frailty (*r* = −0.34, *p* < 0.01) and positively correlated with the DPA (*r* = 0.44, *p* < 0.01), HLSE (*r* = 0.14, *p* < 0.05), and QoL (*r* = 0.46, *p* < 0.01). Results are reported in Table 2.

### 5.3. Testing the Serial Multiple Mediation Model

To test for serial mediation, QoL was entered as the outcome variable and frailty as the predictor variable. DPA and HLSE were entered as serial mediators. In addition, gender, age, and education level, as the most basic and core demographics, significantly correlated with frailty and quality of life; thus, we included these three variables as covariates in the mediation analysis. The result of the serial mediation model for frailty, DPA, HLSE, and QoL revealed a significant negative total effect (coefficient c = −2.99, SE = 0.26, *p* < 0.001). The direct effect of frailty on QoL was proven to be statistically significant (c′ = −1.79, SE = 0.33, *p* < 0.001), indicating that frailty has a significant negative impact on QoL (Table 3).

All three indirect influence paths are significant. The indirect effect M1, the effect of frailty on QoL through DPA (frailty → DPA → QoL), was −0.71 (a1 × b1). This indirect effect was significantly negative because the bootstrap CI did not include zero (CI: −1.14, −0.32). The partial mediating effect of DPA on the relationship between frailty and QoL was therefore significant.

The indirect effect M2, the effect of frailty on QoL through HLSE (frailty → HLSE → QoL), was −0.32 (a2 × b2). This indirect effect was significantly negative because the bootstrap CI did not include zero (CI: −0.59, −0.09). The partial mediating effect of HLSE on the relationship between frailty and QoL was thus significant.

The indirect effect M1M2, the serial effect of frailty on QoL through DPA and HLSE (frailty → DPA → HLSE → QoL), was −0.16 (a1 × d21 × b2). This indirect effect was significantly negative because the bootstrap CI did not include zero (CI: −0.31, −0.04). The serial multiple mediation analysis found a significant link between M1 and M2 (coefficient d21 = 2.65, SE = 0.63, *p* < 0.001).

In addition, DPA and HLSE partially mediated and reduced the impact of frailty on quality of life. After controlling for age, gender, and education level, the regression coefficient of frailty on quality of life (c) was −2.99 (*p*  <  0.001); moreover, three indirect effects of DPA and HLSE and the serial effect of two mediators were added, with the regression coefficient of frailty on quality of life attenuated to −1.79 (c’). The results are shown in Figure 1 and Table 4.

This model includes three indirect effects and one direct effect. The indirect effects are as follows: an indirect effect of frailty on QoL through DPA (M1, a1 × b1), an indirect effect of frailty on QoL through HLSE (M2, a2 × b2), and an indirect effect of frailty on QoL through the serial mediation of DPA and HLSE (M1M2, a1 × d21 × b2). The total effect of frailty (c) is the combination of the direct effect of frailty on QoL and all the indirect effects.

## 6. Discussion

Central to our research findings is the examination in which DPA and HLSE are postulated as jointly mediating variables in the relationship between frailty and QoL in a model. This study is currently one of few studies to focus on frailty and QoL in older adults in Taiwan. The results support the serial mediating effects of DPA and HLSE on the relationship between frailty and QoL. DPA and HLSE may continuously mediate the association between frailty and QoL. Frailty was negatively correlated with DPA, as well as with HLSE, and reducing the negative impact of frailty on QoL was achieved through the mediating effect of DPA and HLSE.

The activity theory of aging describes a positive relationship between a person’s level of activity and life satisfaction [13]. In addition, studies also found physical activity to be positively linked to QoL in older adults [11,41]. Furthermore, DPA can preserve or improve the functioning of many physiological systems in frail older adults [42]. DPA can therefore mediate the influence of frailty on the QoL of older adults. The results from the present study also indicate that frailty predicts QoL directly and indirectly [3,43,44], and that DPA can partially mediate the negative influence of frailty on QoL in older adults. Based on social cognitive theory, self-efficacy is a person’s belief in their capability to successfully perform a particular task [16,17], and it serves an essential function in personal health [16]. Other studies have also found that health-related self-efficacy reflects an individual’s confidence in maintaining their health and influences QoL [19,45].

Another study tracking frailty in stroke patients showed that low self-efficacy increases the progression of frailty and self-efficacy predicts the level of frailty [46]. Studies have shown that physical activity, self-efficacy, and frailty are correlated. However, there may be research evidence–practice gap in frailty management. Regarding care of the elderly, the results of this study show that if older adults in the community can regularly perform activities related to daily living, not only can they improve their self-efficacy toward maintaining their health, but, more importantly, they can also reduce the impact of frailty on their health and quality of life. The results can be used to reduce frailty and improve the quality of life of older adults.

Pender’s health promotion model has been widely adopted to explore the relationships between health promotion behaviors and quality of life [47,48]. By using Pender’s model, this present study further confirms the relationship between daily physical activity (as a prior relevant behavior), self-efficacy for healthy living (as a perception and influence on specific behaviors), and quality of life (as a consequence of health-promoting behaviors). The results of this study support that physical activity behavior and self-efficacy are both important variables in health promotion, and they can reduce the negative impacts of frailty on the quality of life of older adults.

Frailty has become a global health problem and affects healthcare interventions and resources. In this study, the correlation between frailty and DPA (−0.57) was virtually identical to the correlation between frailty and QoL (−0.59). This result is similar to other study findings that have shown that DPA is associated with frailty [49], and that low physical activity is a mediator on the pathway from frailty to activity limitation [50]. In addition, the results of this study are also supported by another study showing that an increased amount of activities relating to daily living can reduce frailty, suggesting that the progression of frailty in older adults can be slowed by maintaining regular activities of daily living [49]. Therefore, previous studies support the results of this study that suggest that DPA may partially mediate the relationship between frailty and QoL.

When the elderly face changes related to aging or frailty, if they can continue to perform their activities of daily living, they should be able improve their debilitating physical condition, improve self-efficacy and confidence in maintaining health, and change the negative impact of aging on their quality of life. The advantage of serial mediation models is that the parallel mediation model assumes that no mediator will causally affect another mediator, whereas in sequential mediation there can be a test for a specific theoretical sequence between variables [25]. Our study validated an important chain relationship between DPA and HLSE (M1→M2), which can reduce the negative impact of frailty on QoL.

The current research had some limitations. First, the use of cross-sectional designs to collect data limits causal inference [51]. Therefore, the results of this study cannot be inferred to elderly people with dependent functions in the community. We also need a large sample to verify the mediating effect of DPA and HLSE on the relationship between frailty and QoL. In addition, longitudinal research is needed to better validate our mediation model in the future [52]. Nevertheless, cross-sectional relationships based on theory and supported by empirical research can still provide valuable information. Moreover, more complicated models and longitudinal research to examine these distinguished relationships are expected in the future [51,52].

## 7. Conclusions

The results of this study demonstrate that frailty negatively impacts quality of life; moreover, DPA, HLSE, and the serial mediation of DPA and HLSE can reduce the negative relationship between frailty and QoL among community-dwelling older adults. According to the results of this study, older adults should be encouraged to continue their activities of daily living, which can not only improve self-efficacy and confidence in maintaining health, but also reduce the negative impact of frailty on quality of life.

## Figures and Tables

**Figure 1 geriatrics-07-00125-f001:**
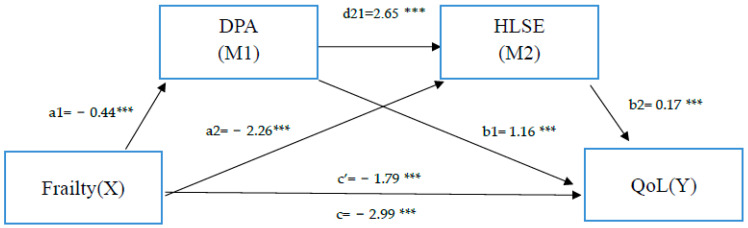
A model of the effect of frailty (independent variable, X), through the mediators DPA (first mediator, M1) and HLSE (second mediator, M2) in series, on QoL (dependent variable, Y). DPA = daily physical activity; HSLE = healthy life self-efficacy; QoL = quality of life; *B* = unstandardized regression coefficients. *** *p* < 0.001.

**Table 1 geriatrics-07-00125-t001:** Descriptive analysis and differences in frailty score, DPA, HLSE, and quality of life with different sociodemographic characteristics (N = 210).

Variables		Frailty Score	DPA	HLSE	QoL
	*n* (%)	Mean ± SD	Mean ± SD	Mean ± SD	Mean ± SD
Total		5.69 ± 3.22	5.92 ± 2.23	75.20 ± 20.76	70.18 ± 15.64
**Gender**					
Male	51 (24%)	4.65 ± 3.27	6.21 ± 1.84	74.69 ± 22.4	73.51 ± 12.58
Female	159 (76%)	6.02 ± 3.14	5.83 ± 2.34	75.37 ± 20.28	69.11 ± 16.39
Statistical Analysis		*t* = −2.63	*t* = 1.21	*t* = −0.19	*t* = 1.75
*p* value		0.01	0.23	0.84	0.08
**Age**					
60–69	98 (47%)	4.70 ± 0.29	1.68 ± 0.17	77.11 ± 20.11	74.71 ± 13.56
70–79	68 (32%)	2.97 ± 0.36	1.84 ± 0.22	75.53 ± 20.22	69.72 ± 14.94
≥80	44 (21%)	3.28 ± 0.50	2.60 ± 0.39	70.45 ± 22.68	60.80 ± 17.01
Statistical Analysis		*F* = 15.48	*F* = 32.33	*F* = 1.58	*F* = 13.51
*p* value		0.001	0.001	0.21	0.001
Marital status					
Unmarried	8 (3%)	6.50 ± 3.25	5.75 ± 2.12	68.38 ± 12.53	71.88 ± 12.80
Married	208 (97%)	5.65 ± 3.22	5.93 ± 2.24	75.48 ± 20.99	70.11 ± 15.77
Statistical Analysis		*F* = 0.73	*F* = −0.23	*F* = −1.52	*F* = 0.38
*p* value		0.49	0.82	0.16	0.72
**Education level**					
≤12 years	178 (85%)	5.94 ± 3.20	5.71 ± 2.27	74.97 ± 21.48	68.57 ± 15.89
≥13 years	32 (15%)	4.28 ± 2.96	7.06 ± 1.58	76.50 ± 16.39	79.13 ± 10.50
Statistical Analysis		*F* = 2.85	*F* = −3.20	*F* = −0.38	*F* = −3.61
*p* value		0.06	0.001	0.70	0.001

Note: *N* = 210; DPA = daily physical activity; HSLE = healthy life self-efficacy; QoL = quality of life.

**Table 2 geriatrics-07-00125-t002:** Correlation matrix for the main variables (*N* = 210).

Variable	M	SD	1	2	3	4	5	6	7
age	74.45	9.15	1						
gender ^a^	-	-	0.10	1					
education level ^a^	-	-	−0.55 **	−0.20 **	1				
frailty	5.69	3.22	0.37 **	0.18 **	−0.34 **	1			
QoL	70.18	15.64	−0.34 **	−0.12	0.46 **	−0.59 **	1		
HLSE	75.20	20.76	−0.09	0.01	0.14 *	−0.62 **	0.51 **	1	
DPA	2.97	2.53	−0.42 **	−0.07	0.44 **	−0.57 **	0.54 **	0.48 **	1

Note: *N* = 210; DPA = daily physical activity; HSLE = healthy life self-efficacy; QoL = quality of life. ** *p* < 0.01, * *p* < 0.05. ^a^ = Spearman correlation coefficient.

**Table 3 geriatrics-07-00125-t003:** Mediating effect of frailty on QoL through DPA and HLSE.

Path	DPA (M1)			HLSE (M2)			QoL (Y)		
		** *B* ** **(LLCI, ULCI)**	*p* value		** *B* ** **(LLCI, ULCI)**	*p* value		** *B* ** **(LLCI, ULCI)**	*p* value
**Frailty (X)**	a1	−0.44 *** (−0.52, −0.36)	<0.001	a2	−2.26 *** (−3.16, −1.36)	<0.001	c’	−1.79 *** (−2.44, −1.14)	<0.001
							c	−2.99 *** (−3.52, −2.47)	<0.001
**DPA (M1)**	-	-	-	d21	2.65 *** (1.41, 3.88)	<0.001	b1	1.61 *** (0.74, 2.49)	<0.001
**HLSE (M2)**	-	-	-	-	-	-	b2	0.14 *** (0.05, 0.24)	<0.001
	R^2^ = 0.60			R^2^ = 0.58			R^2^ = 0.68		
	F = 119.39			F = 53.48			F = 58.91		
	*p* < 0.001			*p* < 0.001			*p* < 0.001		

Note: *N* = 210; DPA = daily physical activity; HSLE = healthy life self-efficacy; QoL = quality of life; CI = confidence interval; SE = standard error; X = independent variable; M1 = first mediator; M2 = second mediator; Y = dependent variable; LLCI = lower limit confidence interval; ULCI = upper limit confidence interval. a, b, c’, and c are the unstandardized coefficients of the path. *** *p* < 0.001.

**Table 4 geriatrics-07-00125-t004:** Bootstrapping indirect effects and 95% confidence intervals (CI) for the final mediation model.

Path	Effect (*B*)	Bootstrap SE	95% CI	
			Bootstrap LLCI	Bootstrap ULCI
Total effect (c)				
frailty → QoL	−2.99	0.26	−3.52	−2.47
Specific indirect effects				
frailty → DPA → QoL(M1)	−0.71	0.20	−1.14	−0.32
frailty → HLSE → QoL(M2)	−0.32	0.13	−0.59	−0.09
frailty → DPA → HLSE → QoL(M1M2)	−0.16	0.06	−0.31	−0.04
Direct effect (c’)				
frailty → QoL	−1.79	0.33	−2.44	−1.14

Note: *N* = 210; Effect = unstandardized coefficient; DPA = daily physical activity; HLSE = healthy life self-efficacy; QoL = quality of life; SE = standard error; LLCI = lower limit confidence interval; ULCI = upper limit confidence interval.

## Data Availability

For original data, please contact the corresponding author; ethical approval does not cover making data openly accessible.

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
