# Peer review of "Frailty and Quality of Life among Older Adults in Communities: The Mediation Effects of Daily Physical Activity and Healthy Life Self-Efficacy"

_geriatrics, 2022, doi:10.3390/geriatrics7060125_

Round 1
Reviewer 1 Report
Dear editor,
Thank you for giving me a good chance to review this interesting manuscript. I have some comments as follows;
1. Abstract:
a. There were some grammatical errors.
b. The authors should add the methodology of this study.
2. Introduction
a. There were some grammatical errors.
b. Please add the benefit based on the aim of this study.
3. Methods
a. This study included older adults aged 60 years or older initially, but the sentence later included the participant's age over 60 years. Please revise them in the same way.
b. Has the EuroQoL visual analog scale (EQ-VAS) been validated in Taiwanese? What is the validity and reliability of this population?
4. Results
a. Participant characteristics: the cutoff point for the degree of frailty and its interpretation according to Lin’s study (2021) should be mentioned in the methods (frailty section).
5. Discussion
a. There was some typo.
b. Please add more explanation about the relationship both direct and indirect among frailty, DPA, self-efficacy, and QoL. For example; Pender's health promotion model.
6. Conclusion
a. The authors should add a brief summary of the relationship between frailty and QOL and the serial mediating effects of the DPA and HLSE
Author Response
Dear Reviewer:
Thanks for providing us with this great opportunity to submit a revised version of our manuscript. We appreciate the detailed and constructive comments provided by the reviewers. We have carefully revised the manuscript by incorporating all the suggestions from the reviewers’ comments.
We hope this revised manuscript has addressed your concerns and look forward to hearing from you.
Sincerely,
Chia-Hui Lin
School of Nursing, Chang Gung University of Science and Technology, Chiayi 61363, Taiwan,
E-Mail: clh9031@gmail.com
Address: No. 8, Sec. W., Jiapu Rd., Puzi City, Chiayi County 613016, Taiwan (R.O.C.)
Encl. Responses to the comments from Reviewers 1

Reviewer 2 Report
The authors have explored the mediating effect of DPA and HLSE on the relationship between frailty and QoL in a sample of older adults (N=210). The analyses and results are uncomplicated, and the authors conclude partial mediating effect of DPA on QoL and HLSE on QoL. Further, series effect of frailty to DPA to HLSE to QoL is reported to be stronger than the either of the simple mediating effect of DPA or HLSE.
Some minor points for revision:
The description about coefficients of the mediation model in statistical section (lines 160-163) can be referred to the figure 1.
The gender distribution in this convenient sample seems to be highly skewed. It would be useful to provide the summary of the variables including age, used in the analysis between male and female sample in a descriptive table (correlation matrix and sample description table can be separate). In this regard correlation matrix can be adjusted for age and sex (partial correlations).
It is not clear age and sex were significantly associated with frailty or QoL. Due to higher percentage of female samples, the authors can speculate about the mediation effect in a stratified analysis.
The HLSE score is mentioned to range between 0 to 10 in the methods section but the mean value in the table one is 75.20. Please clarify.
The covariates used in the analysis can be mentioned in the methods section. To be consistent with the discussion in the text, use lower case for C and C’ used in figure 1.
In the results section, clarify how the series effect is stronger than the individual mediating effect of DPA or HLSE. The actual p-values for the results in the Table 2 can be included (although it is indicated with multiple stars in figure 1).
Author Response
Dear Reviewer:
Thanks for providing us with this great opportunity to submit a revised version of our manuscript. We appreciate the detailed and constructive comments provided by the reviewers. We have carefully revised the manuscript by incorporating all the suggestions from the reviewers’ comments.
We hope this revised manuscript has addressed your concerns and look forward to hearing from you.
Sincerely,
Chia-Hui Lin
School of Nursing, Chang Gung University of Science and Technology, Chiayi 61363, Taiwan,
E-Mail: clh9031@gmail.com
Address: No. 8, Sec. W., Jiapu Rd., Puzi City, Chiayi County 613016, Taiwan (R.O.C.)
Encl. Responses to the comments from Reviewers 2
